# Vision Transformers provably learn spatial structure

**Samy Jelassi**
Princeton University
sjelassi@princeton.edu

**Michael E. Sander**
ENS, CNRS
michael.sander@ens.fr

**Yuanzhi Li**
Carnegie Mellon University
yuanzhil@andrew.cmu.edu

## Abstract

Vision Transformers (ViTs) have achieved comparable or superior performance than Convolutional Neural Networks (CNNs) in computer vision. This empirical breakthrough is even more remarkable since, in contrast to CNNs, ViTs do not embed any visual inductive bias of spatial locality. Yet, recent works have shown that while minimizing their training loss, ViTs specifically learn spatially localized patterns. This raises a central question: how do ViTs learn these patterns by solely minimizing their training loss using gradient-based methods from *random initialization*? In this paper, we provide some theoretical justification of this phenomenon. We propose a spatially structured dataset and a simplified ViT model. In this model, the attention matrix solely depends on the positional encodings. We call this mechanism the positional attention mechanism. On the theoretical side, we consider a binary classification task and show that while the learning problem admits multiple solutions that generalize, our model implicitly learns the spatial structure of the dataset while generalizing: we call this phenomenon patch association. We prove that patch association helps to sample-efficiently transfer to downstream datasets that share the same structure as the pre-training one but differ in the features. Lastly, we empirically verify that a ViT with positional attention performs similarly to the original one on CIFAR-10/100, SVHN and ImageNet.

## 1 Introduction

Transformers are deep learning models built on self-attention [65], and in the past several years they have increasingly formed the backbone for state-of-the-art models in domains ranging from Natural Language Processing (NLP) [65, 23] to computer vision [24], reinforcement learning [13, 38], program synthesis [5] and symbolic tasks [44]. Beyond their remarkable performance, several works reported the ability of transformers to simultaneously minimize their training loss and learn inductive biases tailored to specific datasets e.g. in computer vision [55], in NLP [10, 67] or in mathematical reasoning [73]. In this paper, we focus on computer vision where convolutions are considered to be an adequate and biologically plausible inductive bias since they capture local spatial information [27] by imposing a sparse local connectivity pattern. This seems intuitively reasonable: nearby pixels encode the presence of small scale features, whose patterns in turn determine more abstract features at longer and longer length scales. Several seminal works [17, 24, 55] *empirically* show that although randomly initialized, the positional encodings in Vision transformers (ViTs) [24] actually learn this local connectivity: closer patches have more similar positional encodings, as shown in Figure 1a. A priori, learning such spatial structure is surprising. Indeed, in contrast to convolutional neural networks (CNNs), ViTs are not built with the inductive bias of local connectivity and weight sharing. They start by replacing an image by a collection of $D$ patches $(\boldsymbol{X}_1, \ldots, \boldsymbol{X}_D) \in \mathbb{R}^{d \times D}$, each of dimension $d$. While each $\boldsymbol{X}_i$ represents (an embedding of) a spatially localized portion of the original image, the relative positions of the patches $\boldsymbol{X}_i$ in the image are disregarded. Instead, relative spatial information is supplied through *image-independent* positional encodings $\boldsymbol{P} = (\boldsymbol{p}_1, \ldots, \boldsymbol{p}_D) \in \mathbb{R}^{d \times D}$. Unlike CNNs, each layer of a ViT then learns, via trainable self-attention, a non-local set of filters that non-linearly depend on both the values of all patches $\boldsymbol{X}_j$ and their positional encodings $\boldsymbol{p}_j$.

36th Conference on Neural Information Processing Systems (NeurIPS 2022).

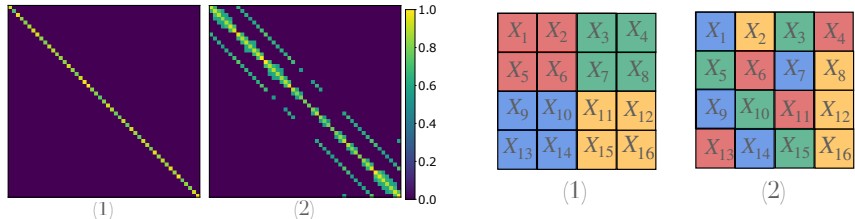

(a)                      (b)

Figure 1: (a) Visualization of the positional encodings similarities $\boldsymbol{P}^\top \boldsymbol{P} = (\langle \boldsymbol{p}_i, \boldsymbol{p}_j \rangle)_{(i,j) \in [D]^2}$ at initialization (1) and after training on Imagenet (2) using a "ViT-small-patch32-224" [24]. We normalise the values $\boldsymbol{P}^\top \boldsymbol{P}$ between $-1$ and $1$ and apply a threshold of $0.55$. In contrast with the initial arrays that are random, the final ones show local connectivity patterns: nearby patches have similar positional encodings. (b) Partition of the patches into sets $\mathcal{S}_\ell$ as in Definition 2.1. Squares in the same color belong to the same set $\mathcal{S}_\ell$. We refer to (1) as a "spatially localized set" since all the elements in a $\mathcal{S}_\ell$ are spatially contiguous. This is the type of sets appearing in Figure 1a at the end of training. Definition 2.1 also covers sets with non-contiguous elements as (2).

**Contributions.** The empirical observation of Figure 1a sets a central question: from a *theoretical perspective*, how do ViTs manage to learn these local connectivity patterns by simply minimizing their training loss using gradient descent from random initialization? While it is known that attention can express local operations as convolution [17], it remains unclear how ViTs learn it. In this paper, we present a simple spatially-structured classification dataset for which it is sufficient (but not necessary) to learn the structure in order to generalize. We also present a simplified ViT model which we prove implicitly learns sparse spatial connectivity patterns when it minimizes its training loss via gradient descent (GD). We name this implicit bias *patch association* (defined in Definition 2.2). We prove that our ViT model leverages this bias to generalize. More precisely, we make the following contributions:

– In Section 2, we formally define the concept of performing patch association, which refer to the ability of learning spatial connectivity patterns on a dataset.
– In Section 3, we introduce a structured classification dataset and a simplified ViT model. This model is simplified in the sense that its attention matrix only depends on the positional encodings. We then present the learning problems we are interested in: empirical risk (realistic setting) and population risk (idealized setting) minimization for binary classification.
– In Section 4, we prove that a one-layer single-head ViT model trained with gradient descent on our synthetic dataset performs patch association and generalizes, in the idealized (Theorem 4.1) and realistic (Theorem 4.2) settings. We present a detailed proof, based on invariance and symmetries of coefficients in the attention matrix throughout the learning process.
– In Section 5, we show (Theorem 5.1) that after pre-training in our synthetic dataset, our model can be sample-efficiently fine-tuned to transfer to a downstream dataset that shares the same structure as the source dataset (and may have different features).
– On the experimental side, we validate in Section 6 that ViTs learn spatial structure in images from the CIFAR-100 dataset, even when the pixels of the images are permuted. This result validates that, in contrast to CNNs, ViTs learn a more general form of spatial structure that is not limited to local patterns (Figure 5). We finally show that our ViT model –where the attention matrix only depends on the positional encodings– is competitive with the vanilla ViT on the ImageNet, CIFAR-10/100 and SVHNs datasets (Figure 6 and Figure 7).

**Notation.** We use lower case letters for scalars, lower case bold for vectors and upper case bold for matrices. Given an integer $D$, we define $[D] = \{1, \ldots, D\}$. Any statement made "with high probability" holds with probability at least $1 - 1/\mathrm{poly}(d)$. Given a vector $\boldsymbol{a} \in \mathbb{R}^d$ and $k \leqslant d$, we define $\mathrm{Top}_k \{a_j\}_{j=1}^d = \{a_{i_1}, \ldots, a_{i_k}\}$ where $a_{i_1}, \ldots, a_{i_k}$ are the $k$-largest elements. For a function $F$ that implicitly depend on parameters $\boldsymbol{A}$ and $\boldsymbol{v}$, we often write $F_{\boldsymbol{A},\boldsymbol{v}}$ to highlight its parameters. We use the asymptotic complexity notations when defining the different constants.

## Related work

**CNNs and ViTs.** Many computer vision architectures can be considered as a form of hybridization between Transformers and CNNs. For example, DeTR [11] use a CNN to generate features that are fed to a Transformer. [25] show that self-attention can be initialized or regularized to behave like a convolution and [19, 30] add convolution operations to Transformers. Conversely, [8, 56, 7] introduce self-attention or attention-like operations to supplement or replace convolution in ResNet-

like models. In contrast, our paper does not consider any form of hybridization with CNN, but rather a simplification of the original ViT to explain how ViTs learn spatially structured patterns using GD.

**Empirical understanding of ViTs.** A long line of work consists in analyzing the properties of ViTs, such as robustness [9, 54, 51] or the effect of self-supervision [12, 14]. Closer to our work, some papers investigate why ViTs perform so well. [55] compare the representations of ViTs and CNNs and [50, 64] argue that the patch embeddings could explain the performance of ViTs. We empirically show in Section 6 that applying the attention matrices to the positional encodings – which contains the structure of the dataset – approximately recovers the baselines. Hence, our work rather suggests that the structural learning performed by the attention matrices may explain the success of ViTs.

**Theory for attention models.** Early theoretical works have focused on the expressivity of attention. [66, 26] addressed this question in the context of self-attention blocks and [21, 68, 34] for Transformers. On the optimization side, [76] investigate the role of adaptive methods in attention models and [59] analyze the dynamics of a single-head attention head to approximate the learning of a Seq2Seq architecture. In our work, we also consider a single-head ViT trained with gradient descent and exhibit a setting where it provably learns convolution-like patterns and generalizes.

**Algorithmic regularization.** The question we address concerns algorithmic regularization which characterizes the generalization of an optimization algorithm when multiple global solutions exist in over-parametrized models. This regularization arises in deep learning mainly due to the *non-convexity* of the objective function. Indeed, this latter potentially creates multiple global minima scattered in the space that vastly differ in terms of generalization. Algorithmic regularization appears in binary classification [60, 49, 16], matrix factorization [29, 3], convolutional neural networks [29, 37], generative adversarial networks [2], contrastive learning [70] and mixture of experts [15]. Algorithmic regularization is induced by and depends on many factors such as learning rate and batch size [28, 33, 41, 58, 47], initialization [1], momentum [39], adaptive step-size [42, 53, 20, 71, 78, 40], batch normalization [4, 32, 36] and dropout [61, 69]. However, all these works consider the case of feed-forward neural networks which does not apply to ViTs.

## 2 Defining patch association

The goal of this section is to formalize the way ViTs learn sparse spatial connectivity patterns. We thus introduce the concept of performing patch association for a spatially structured dataset.

**Definition 2.1** (Data distribution with spatial structure). *Let $\mathcal{D}$ be a distribution over $\mathbb{R}^{d \times D} \times \{-1, 1\}$ where each patch $\boldsymbol{X} = (\boldsymbol{X}_1, \dots, \boldsymbol{X}_D) \in \mathbb{R}^{d \times D}$ has label $y \in \{-1, 1\}$. We say that $\mathcal{D}$ is spatially structured if*

– *there exists a partition of $[D]$ into $L$ disjoint subsets i.e. $[D] = \bigcup_{\ell=1}^{L} \mathcal{S}_\ell$ with $\mathcal{S}_\ell \subsetneq D$ and $|\mathcal{S}_\ell| = C$.*

– *there exists a labeling function $f^*$ satisfying $\mathbb{P}[yf^*(\boldsymbol{X}) > 0] = 1 - d^{-\omega(1)}$ and,*

$$f^*(\boldsymbol{X}) := \sum_{\ell \in [L]} \phi((\boldsymbol{X}_i)_{i \in \mathcal{S}_\ell}), \quad \text{where } \phi \colon \mathbb{R}^{d \times C} \to \mathbb{R} \text{ is an arbitrary function.} \tag{1}$$

**Examples.** A particular case for the sets $\mathcal{S}_\ell$'s is the one of spatially localized sets as in Figure 1b-(1). In this case, we have $D = 16$, $C = 4$ and $\mathcal{S}_1 = \{1, 2, 5, 6\}$, $\mathcal{S}_2 = \{3, 4, 7, 8\}$, $\mathcal{S}_3 = \{9, 10, 13, 14\}$, $\mathcal{S}_4 = \{11, 12, 15, 16\}$. We emphasize that Definition 2.1 is not limited to spatially localized sets and also covers non-contiguous sets as Figure 1b-(2).

**Labelling function** Definition 2.1 states that there exists a labelling function that preserves the underlying structure by applying the same function $\phi$ to each $\mathcal{S}_\ell$'s as in (1). For instance, when the sets $\mathcal{S}_\ell$'s

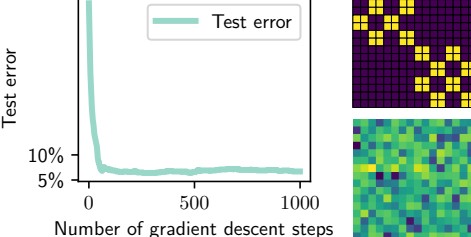

Figure 2: Left: Test error of the ViT on the convolution structured dataset. Upper Right: Grid displaying the input patches. Yellow squares represent spatially localized sets $\mathcal{S}_\ell$. Those sets are taken into account when computing the convolutional function $f^*$. Lower Right: Learnt $\boldsymbol{P}^\top \boldsymbol{P}$ looks random compared to upper one.

are spatially localized, $f^*$ can be a one-hidden layer convolutional network. In this paper, we are interested in *patch association* which refers to the ability of an algorithm to identify the sets $\mathcal{S}_\ell$'s, and is formally defined as follow.

**Definition 2.2** (Patch association for ViTs). *Let $\mathcal{D}$ be as in Definition 2.1. Let $\mathcal{M}: \mathbb{R}^{d \times D} \to \{-1, 1\}$ be a transformer and $\boldsymbol{P}^{(\mathcal{M})}$ its positional encodings matrix. We say that $\mathcal{M}$ performs patch association on $\mathcal{D}$ if for all $\ell \in [L]$ and $i \in \mathcal{S}_\ell$, we have $\text{Top}_C \{\langle \boldsymbol{p}_i^{(\mathcal{M})}, \boldsymbol{p}_j^{(\mathcal{M})} \rangle\}_{j=1}^D = \mathcal{S}_\ell$.*

Definition 2.2 states that patch association is learned when for a given $i \in \mathcal{S}_\ell$, its positional encoding mainly attends those of $j$ such that $i, j \in \mathcal{S}_\ell$. In this way, the transformer groups the $\boldsymbol{X}_i$ according to $\mathcal{S}_\ell$ just like the true labeling function. Definition 2.2 formally describes the empirical findings in Figure 1a-(2), where nearby patches have similar positional encodings. A natural question is then: would ViTs really learn those $\mathcal{S}_\ell$ after training to match the labeling function $f^*$? Without further assumptions on the data distribution, we next show that the answer is no.

**ViTs do not always learn patch association under Assumption 1.** We give a negative answer through the following synthetic experiment. Consider the case where all the patches $\boldsymbol{X}_j$ are i.i.d. standard Gaussian and $f^*$ is a one-hidden layer CNN with cubic activation. The label $y$ of any $\boldsymbol{X}$ is then given by $y = \text{sign}(f^*(\boldsymbol{X}))$. As shown in Figure 2, one-layer ViT reaches small test error on the binary classification task. However, $\boldsymbol{P}^\top \boldsymbol{P}$ does not match the convolution pattern encoded in $f^*$. This is not surprising, since the data distribution $\mathcal{D}$ is Gaussian, and thus lacks spatial structure. Thus, in order to prove that ViTs learn patch association, we need additional assumptions on $\mathcal{D}$, which we discuss in the next section.

## 3 Setting to learn patch association

In this section, we introduce our theoretical setting to analyze how ViTs learn patch association. We first define our binary classification dataset and finally present the ViT model we use to classify it.

**Assumption 1** (Data distribution with specific spatial structure). *Let $\mathcal{D}$ be a distribution as in Definition 2.1 and $\boldsymbol{w}^* \in \mathbb{R}^d$ be an underlying feature. We suppose that each data-point $\boldsymbol{X}$ is defined as follow*

– *Uniformly sample an index $\ell(\boldsymbol{X})$ from $[L]$ and for $j \in \mathcal{S}_{\ell(\boldsymbol{X})}$, $\boldsymbol{X}_j = y\boldsymbol{w}^* + \boldsymbol{\xi}_j$, where $y\boldsymbol{w}^*$ is the informative feature and $\boldsymbol{\xi}_j \overset{i.i.d.}{\sim} \mathcal{N}(0, \sigma^2(\mathbf{I}_D - \boldsymbol{w}^*\boldsymbol{w}^{*\top}))$ (signal set).*

– *For $\ell \in [L]\backslash\{\ell(\boldsymbol{X})\}$ and $j \in \mathcal{S}_\ell$, $\boldsymbol{X}_j = \delta_j \boldsymbol{w}^* + \boldsymbol{\xi}_j$, where $\delta_j = 1$ with probability $q/2$, $-1$ with same probability and $0$ otherwise, and $\boldsymbol{\xi}_j \overset{i.i.d.}{\sim} \mathcal{N}(0, \sigma^2(\mathbf{I}_D - \boldsymbol{w}^*\boldsymbol{w}^{*\top}))$ (random sets).*

To keep the analysis simple, the noisy patches are sampled from the orthogonal complement of $\boldsymbol{w}^*$. Note that $\mathcal{D}$ admits the labeling function $f^*(\boldsymbol{X}) = \sum_{\ell \in [L]} \text{Threshold}_{0.9C}(\sum_{i \in \mathcal{S}_\ell}\langle \boldsymbol{w}^*, \boldsymbol{X}_i \rangle)$, where $\text{Threshold}_C(z) = z$ if $|z| > C$ and $0$ otherwise.

We sketch a data-point of $\mathcal{D}$ in Figure 3. Our dataset can be viewed as an extreme simplification of real-world image datasets where there is a set of adjacent patches that contain a useful feature (e.g. the nose of a dog) and many patches that have uninformative or spurious features e.g. the background of the image. We make the following assumption on the parameters of the data distribution.

**Assumption 2.** *We suppose that $d = \text{poly}(D)$, $C = \text{polylog}(d)$, $q = \text{poly}(C)/D$, $\|\boldsymbol{w}^*\|_2 = 1$ and $\sigma^2 = 1/d$. This implies $C \ll D$ and $q \ll 1$.*

Assumption 2 may be justified by considering a "ViT-base-patch16-224" model [24] on ImageNet. In this case, $d = 384$, $D = 196$. $\sigma$ is set to have $\|\boldsymbol{\xi}_j\|_2 \approx \|\boldsymbol{w}^*\|_2$. $q$ is chosen so that there are more spurious features than informative ones

Figure 3: Visualization of a data-point $\boldsymbol{X}$ in $\mathcal{D}$ when the $\mathcal{S}_\ell$'s are spatially localized. Each square depicts a patch $\boldsymbol{X}_j$ and squares of the same color belong to the same set $\mathcal{S}_\ell$. "0" indicates that the patch does not have a feature, "1" stands for feature $1 \cdot \boldsymbol{w}^*$ and "-1" for feature $-1 \cdot \boldsymbol{w}^*$. The large red square depicts the signal set $\ell(\boldsymbol{X})$. Although there are more "-1"'s than "+1"'s, the label of $\boldsymbol{X}$ is +1 since there are only "+1"'s inside the signal set.

(low signal-to-noise regime) which makes the data *non-linearly separable*. Our dataset is non-trivial to learn since generalized linear networks fail to generalize, as shown in the next theorem (see Appendix J for a proof).

**Theorem 3.1.** *Let $\mathcal{D}$ be as in Assumption 1. Let $g(\boldsymbol{X}) = \phi\left(\sum_{j=1}^{D}\langle \boldsymbol{w}_j, \boldsymbol{X}_j\rangle\right)$ be a generalized linear model. Then, $g$ does not fit the labeling function i.e. $\mathbb{P}[f^*(\boldsymbol{X})g(\boldsymbol{X}) \leqslant 0] \geqslant 1/8$.*

Intuitively, $g$ fails to generalize because it does not have any knowledge on the underlying partition and the number of random sets is much higher than those with signal. Thus, a model must have a minimal knowledge about the $\mathcal{S}_\ell$'s in order to generalize. In addition, the following Theorem 3.2 states the existence of a transformer that generalizes without learning spatial structure (see Appendix J for a proof), thus showing that the learning process has a priori no straightforward reason to lead to patch association.

**Theorem 3.2.** *Let $\mathcal{D}$ be defined as in Assumption 1. There exists a (one-layer) transformer $\mathcal{M}$ so that $\mathbb{P}[f^*(\boldsymbol{X})\mathcal{M}(\boldsymbol{X}) \leqslant 0] = d^{-\omega(1)}$ but for all $\ell \in [L]$, $i \in \mathcal{S}_\ell$, $\mathrm{Top}_C\ \{\langle \boldsymbol{p}_i^{(\mathcal{M})}, \boldsymbol{p}_j^{(\mathcal{M})}\rangle\}_{j=1}^{D} \cap \mathcal{S}_\ell = \varnothing$.*

**Simplified ViT model.** We now define our simplified ViT model for which we show in Section 4 that it implicitly learns patch association via minimizing its training objective. We first remind the self-attention mechanism that is ubiquitously used in transformers.

**Definition 3.1** (Self-attention [6, 65])**.** *The attention mechanism [6, 65] in the single-head case is defined as follow. Let $\boldsymbol{X} \in \mathbb{R}^{d \times D}$ a data point and $\boldsymbol{P} \in \mathbb{R}^{d \times D}$ its positional encoding. The self-attention mechanism computes*

1. *the sum of patches and positional encodings i.e. $\boldsymbol{\mathcal{X}} = \boldsymbol{X} + \boldsymbol{P}$.*
2. *the attention matrix $\boldsymbol{A} = \boldsymbol{Q}\boldsymbol{K}^\top$ where $\boldsymbol{Q} = \boldsymbol{\mathcal{X}}^\top \boldsymbol{W_Q}$, $\boldsymbol{K} = \boldsymbol{\mathcal{X}}^\top \boldsymbol{W_K}$, $\boldsymbol{W_Q}, \boldsymbol{W_K} \in \mathbb{R}^{d \times d}$.*
3. *the score matrix $\boldsymbol{S} \in \mathbb{R}^{D \times D}$ with coefficients $S_{i,j} = \exp(A_{i,j}/\sqrt{d})/\sum_{r=1}^{D}\exp(A_{i,r}/\sqrt{d})$.*
4. *the matrix $\boldsymbol{V} = \boldsymbol{\mathcal{X}}^\top \boldsymbol{W_V}$, where $\boldsymbol{W_V} \in \mathbb{R}^{d \times d}$.*

*It finally outputs $\mathrm{SA}((\boldsymbol{X}; \boldsymbol{P})) = \boldsymbol{S}\boldsymbol{V} \in \mathbb{R}^{d \times D}$.*

In this paper, our ViT model relies on a different attention mechanism –the "positional attention"– that we define as follows.

**Definition 3.2** (Positional attention)**.** *Let $\boldsymbol{X} \in \mathbb{R}^{d \times D}$ and $\boldsymbol{P} \in \mathbb{R}^{d \times D}$ the positional encoding. The positional attention mechanism takes as input the pair $(\boldsymbol{X}; \boldsymbol{P})$ and computes:*

1. *the attention matrix $\boldsymbol{A} = \boldsymbol{Q}\boldsymbol{K}^\top$ where $\boldsymbol{Q} = \boldsymbol{P}^\top \boldsymbol{W_Q}$, $\boldsymbol{K} = \boldsymbol{P}^\top \boldsymbol{W_K}$ and $\boldsymbol{W_Q}, \boldsymbol{W_K} \in \mathbb{R}^{d \times d}$.*
2. *the score matrix $\boldsymbol{S} \in \mathbb{R}^{D \times D}$ with coefficients $S_{i,j} = \exp(A_{i,j}/\sqrt{d})/\sum_{r=1}^{D}\exp(A_{i,r}/\sqrt{d})$.*
3. *the matrix $\boldsymbol{V} = \boldsymbol{X}^\top \boldsymbol{W_V}$, where $\boldsymbol{W_V} \in \mathbb{R}^{d \times d}$.*

*It outputs $\mathrm{PA}((\boldsymbol{X}; \boldsymbol{P})) = \boldsymbol{S}\boldsymbol{V}$.*

Positional attention *isolates* positional encoding $\boldsymbol{P}$ from data $\boldsymbol{X}$: $\boldsymbol{A}$ encodes the dynamics of $\boldsymbol{P}$ and tracks whether patch association is learned. $\boldsymbol{V}$ encodes the data-dependent part and monitors whether the feature is learned. Indeed, given its highly non-linear nature with respect to the input, directly analyzing self-attention is difficult. Yet, positional attention is similar to self-attention. As this latter, positional attention is also permutation-invariant and processes all tokens simultaneously. Besides, positional attention also computes a score matrix between the different tokens. This similarity matrix is also normalized in a sparse manner with the Softmax operator. The only aspect that positional attention misses from self-attention is the fact that $\boldsymbol{S}$ does not depend on the input. Nevertheless, we empirically show that our positional attention model competes with self-attention in Section 6. Lastly, we make the following simplification in the parameters to ease our analysis.

**Simplification 3.1.** *In the positional attention mechanism, we set $d = D$, $\boldsymbol{W_K} = \boldsymbol{I}_D$ and $\boldsymbol{W_Q} = \boldsymbol{I}_D$ which implies $\boldsymbol{A} = \boldsymbol{P}^\top \boldsymbol{P}$. We set $\boldsymbol{W_V} = [\boldsymbol{v}, \ldots, \boldsymbol{v}] \in \mathbb{R}^{d \times D}$ where $\boldsymbol{v} \in \mathbb{R}^d$. Finally, we set $\boldsymbol{A}$ and $\boldsymbol{v}$ as trainable parameters. Besides, without loss of generality, we train all $A_{i,j}$ for $i \neq j$ and leave the diagonals of $\boldsymbol{A}$ fixed.*

In Simplification 3.1, we set $\boldsymbol{W_K}$ and $\boldsymbol{W_Q}$ to the identity so that $\boldsymbol{A} = \boldsymbol{P}^\top \boldsymbol{P}$. This Gram matrix encodes the spatial patterns learned by the ViT as shown in Figure 1a. Besides, since fitting the

labeling function requires to learn one feature $\boldsymbol{w}^*$, it is sufficient to parameterize $\boldsymbol{W_V}$ with a vector $\boldsymbol{v}$. Also, although $\boldsymbol{A} = \boldsymbol{P}^\top \boldsymbol{P}$ and $\boldsymbol{P}$ is trainable, we choose for simplicity to only optimize over $\boldsymbol{A}$. Besides, we leave the $A_{i,i}$'s fixed because Softmax is invariant under the uniform shift of the input. Under Simplification 3.1, our simplified ViT model is then a two attention layer with a single head:

$$F(\boldsymbol{X}) = \sum_{i=1}^{D} \sigma\left( D \sum_{j=1}^{D} S_{i,j}\langle \boldsymbol{v}, \boldsymbol{X}_j\rangle \right) \quad \text{with} \quad S_{i,j} = \exp(A_{i,j}/\sqrt{d}) / \sum_{r=1}^{D} \exp(A_{i,r}/\sqrt{d}), \quad \text{(T)}$$

where $\sigma$ is an activation function. Since we aim to the simplest ViT model, we opt for a polynomial activation i.e. $\sigma(x) = x^p + \nu x$ where $p \geqslant 3$ is an odd integer and $\nu = 1/\text{poly}(d)$. Note that this choice of polynomial activation is common in the deep learning theory literature – see e.g. [45, 1, 72] among others. The degree $p$ is odd to make the ViT model compatible with the labeling function and strictly larger than 1 because the data is not linearly separable (Theorem 3.1). We add a linear part in the activation function to ensure that the gradient is non-zero when $\boldsymbol{v}$ has small coefficients. With these simplifications, we formally prove that $F$ is able to learn patch association and generalize, in the two following settings.

**Idealized and realistic learning problems.** Given a dataset $\mathcal{Z} = \{(\boldsymbol{X}[i], y[i])\}_{i=1}^{N}$ sampled from $\mathcal{D}$, we solve the empirical risk minimization problem for the logistic loss defined by:

$$\min_{\widehat{\boldsymbol{A}}, \widehat{\boldsymbol{v}}} \quad \frac{1}{N} \sum_{i=1}^{N} \log\left(1 + e^{-y[i]F(\boldsymbol{X}[i])}\right) := \widehat{\mathcal{L}}(\widehat{\boldsymbol{A}}, \widehat{\boldsymbol{v}}). \quad \text{(E)}$$

Instead of directly analyzing (E), we introduce a proxy where we minimize the population risk

$$\min_{\boldsymbol{A}, \boldsymbol{v}} \quad \mathbb{E}_{\mathcal{D}}\left[\log\left(1 + e^{-yF(\boldsymbol{X})}\right)\right] := \mathcal{L}(\boldsymbol{A}, \boldsymbol{v}). \quad \text{(P)}$$

We refer to (E) as the *realistic* problem while (P) as the *idealized* problem.

**Algorithm.** We solve (P) and (E) using gradient descent (GD) for $T$ iterations. The update rule in the case of (P) for $t \in [T]$ and $i, j \in [D]$ is

$$A_{i,j}^{(t+1)} = A_{i,j}^{(t)} - \eta \partial_{A_{i,j}} \mathcal{L}(\boldsymbol{A}^{(t)}, \boldsymbol{v}^{(t)}), \quad \boldsymbol{v}^{(t+1)} = \boldsymbol{v}^{(t)} - \eta \nabla_{\boldsymbol{v}} \mathcal{L}(\boldsymbol{A}^{(t)}, \boldsymbol{v}^{(t)}), \quad \text{(GD)}$$

where $\eta > 0$ is the learning rate. A similar update may be written for (E). We now detail how to set the parameters in (GD).

**Parametrization 3.1.** *When running GD on (P) and (E), the number of iterations is any $T \geqslant \text{poly}(d)/\eta$. We set the learning rate as $\eta \in \left(0, \frac{1}{\text{poly}(d)}\right)$. The diagonal coefficient of the attention matrix are set for $i \in [D]$ as $A_{i,i}^{(0)} = \widehat{A}_{i,i}^{(0)} = \sigma_{\boldsymbol{A}}\mathbf{I}_D$ where $\sigma_{\boldsymbol{A}} = \text{polyloglog}(d)$. The off-diagonal coefficients of $\boldsymbol{A}$ and the value vector are initialized as:*

1. *Idealized case: $\boldsymbol{v}^{(0)} = \alpha^{(0)}\boldsymbol{w}^*$ where $\alpha^{(0)} = \nu^{1/(p-1)}$ and $A_{i,j}^{(0)} = 0$ for $i \neq j$.*

2. *Realistic case: $\widehat{\boldsymbol{v}}^{(0)} \sim \mathcal{N}(0, \omega^2 \mathbf{I}_d)$ and $\widehat{A}_{i,j}^{(0)} \sim \mathcal{N}(0, \omega^2)$ where $i \neq j$ and $\omega = 1/\text{poly}(d)$.*

We remind that in Simplification 3.1, we have $\boldsymbol{A} = \boldsymbol{P}^\top \boldsymbol{P}$. If one initializes $\boldsymbol{P} \sim \mathcal{N}(0, \sigma_A \mathbf{I}_D/D)$, then with high probability, $A_{i,i}^{(0)} = \|\boldsymbol{p}_i^{(0)}\|_2^2 = \Theta(\sigma_A)$ and $A_{i,j}^{(0)} = \langle \boldsymbol{p}_i^{(0)}, \boldsymbol{p}_j^{(0)}\rangle = \Theta(\sigma_A/\sqrt{D})$ for $i \neq j$. Since $D \gg 1$, it is then reasonable to set $A_{i,j}^{(0)} = 0$. Note that, also in the idealized setting, we initialize $\boldsymbol{v}^{(0)}$ in $\text{span}(\boldsymbol{w}^*)$, even though this latter should be unknown to the algorithm. We remind that the idealized case is a proxy to ultimately characterize the realistic dynamics.

## 4  Learning spatial structure via matching the labeling function

As announced above, we show that our ViT (T) implicitly learns patch association and fits the labeling function by minimizing the training objective. We first study the dynamics in (P). Using the analysis in the idealized case, we then characterize the solution found in the realistic problem (E).

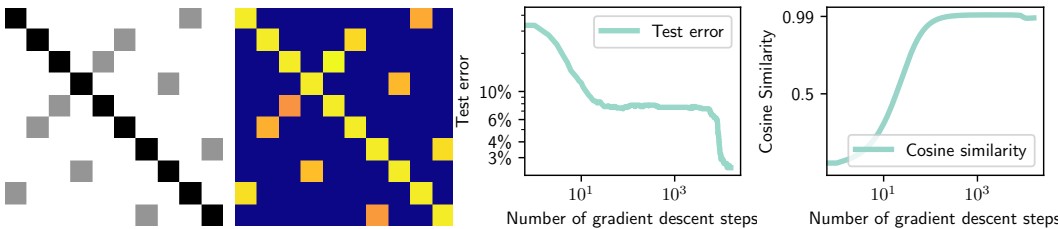

Figure 4: Illustration of Theorem 4.2. We consider the exact same setting (data generation, parameter settings...) as for the realistic case. From left to right, we first display in grey the tuples $(i, j)$ such that $(i, j) \in \mathcal{S}_\ell$. We then plot the learned matrix $\boldsymbol{A}$ and see that coefficients with high value exactly correspond to their grey scale counterpart in the left plot. We also display test error and cosine similarity between $\boldsymbol{w}^*$ and $\boldsymbol{v}$ w.r.t the number of training steps.

## 4.1 Learning process in the idealized case

In this section, we analyze the dynamics of (P). Our main result is that after minimizing (P), our model (T) performs patch association while generalizing.

**Theorem 4.1.** *Assume that we run GD on (P) for $T$ iterations with parameters set as in Parametrization 3.1. With high probability, the ViT model (T)*

1. *learns patch association i.e. for all $\ell \in [L]$ and $i \in \mathcal{S}_\ell$, $\mathrm{Top}_C \{A_{i,j}^{(T)}\}_{j=1}^D = \mathcal{S}_\ell$.*

2. *learns the labeling function $f^*$ i.e. $\mathbb{P}_{\mathcal{D}}[f^*(\boldsymbol{X}) F_{\boldsymbol{A}^{(T)}, \boldsymbol{v}^{(T)}}(\boldsymbol{X}) > 0] \geq 1 - o(1)$.*

We now sketch the main ideas to prove the theorem for which one can refer to Appendix D for a complete proof.

**Invariance and symmetries.** In (P), we take the expectation over $\mathcal{D}$. Since (T) is permutation-invariant and the data distribution is symmetric, we can thus dramatically simplify the variables in (P). An illustration of this is the next lemma that shows that $\boldsymbol{A}$ can be reduced to three variables in (P).

**Lemma 4.1.** *There exist $\beta = \sigma_{\boldsymbol{A}}$, $\gamma^{(t)}, \rho^{(t)} \in \mathbb{R}$ such that for all $t \geq 0$:*

1. *for all $i \in [D]$, $A_{i,i}^{(t)} = \beta$.*

2. *for all $i, j \in [D]$ such that $i, j \in \mathcal{S}_\ell$ for some $\ell \in [L]$, $A_{i,j}^{(t)} = \gamma^{(t)}$.*

3. *for all $i, j \in [D]$ such that $i \in \mathcal{S}_\ell$ and $j \in \mathcal{S}_m$ for some $\ell, m \in [L]$ with $\ell \neq m$, $A_{i,j}^{(t)} = \rho^{(t)}$.*

Besides, using the initialization in Parametrization 3.1, we can show that $\boldsymbol{v}$ always lies in $\mathrm{span}(\boldsymbol{w}^*)$.

**Lemma 4.2.** *For all $t \in [T]$, there exists $\alpha^{(t)} \in \mathbb{R}$ such that $\boldsymbol{v}^{(t)} = \alpha^{(t)} \boldsymbol{w}^*$.*

In summary, Lemma 4.1 and Lemma 4.2 imply that instead of optimizing over $\boldsymbol{A}$ and $\boldsymbol{v}$ in (P), we can instead consider the scalar variables $\alpha^{(t)}$, $\gamma^{(t)}$ and $\rho^{(t)}$. The remaining of this section consists in analyzing the dynamics of these three quantities.

**Learning patch association.** We first analyze the dynamics of $\gamma^{(t)}$ and $\rho^{(t)}$. To this end, we introduce the following terms:

$$\Lambda^{(t)} = \frac{e^\beta}{e^\beta + (C-1)e^{\gamma^{(t)}} + (D-C)e^{\rho^{(t)}}}, \qquad \Gamma^{(t)} = \frac{e^{\gamma^{(t)}}}{e^\beta + (C-1)e^{\gamma^{(t)}} + (D-C)e^{\rho^{(t)}}},$$

$$\Xi^{(t)} = \frac{e^{\rho^{(t)}}}{e^\beta + (C-1)e^{\gamma^{(t)}} + (D-C)e^{\rho^{(t)}}}, \qquad G^{(t)} = D(\Lambda^{(t)} + (C-1)\Gamma^{(t)}).$$

Note that $\Lambda^{(t)}$, $\Gamma^{(t)}$ and $\Xi^{(t)}$ respectively correspond to the coefficients on the diagonal, those for which $i, j \in \mathcal{S}_\ell$ for some $\ell \in [L]$ and all the other coefficients of the attention matrix $\boldsymbol{S}$. Using these notations, we first derive the GD updates of $\gamma^{(t)}$ and $\rho^{(t)}$.

**Lemma 4.3.** *Let $t \leq T$. The attention weights $\gamma^{(t)}$ and $\rho^{(t)}$ satisfy:*

$$\gamma^{(t+1)} = \gamma^{(t)} + \eta \mathrm{polylog}(d)(\alpha^{(t)})^p \cdot \Gamma^{(t)}(G^{(t)})^{p-1},$$

$$|\rho^{(t+1)}| \leq |\rho^{(t)}| + \eta \mathrm{polylog}(d)(\alpha^{(t)})^p \left(\frac{1}{D} + \frac{1}{D}\Gamma^{(t)}(G^{(t)})^{p-1}\right).$$

Lemma 4.3 shows that the increment of $\gamma^{(t)}$ is larger than the one of $\rho^{(t)}$. Since $\gamma^{(0)} = \rho^{(0)} = 0$, this implies that $\gamma^{(t)} \geqslant \rho^{(t)}$ for all $t \geqslant 0$. This observation proves the first item of Theorem 4.1. We now explain how learning patch association leads to $v$ highly correlated with $w^*$.

- **Event I**: At the beginning of the process, the update of $v^{(t)}$ is larger than the one of $A_{i,j}^{(t)}$ which implies that only $v^{(t)}$ updates during this first phase. We show that $\alpha^{(t)} = \langle v^{(t)}, w^* \rangle$ increases until a time $\mathcal{T}_0 > 0$ where it reaches some threshold (Lemma D.2). At this point, the model is nothing else than a generalized linear model that would not generalize because there are much more noisy tokens than signal ones (see Theorem 3.1).

- **Event II**: During this phase, the attention weights must update. Indeed, assume by contradiction that the $A_{i,j}^{(t)}$ stay around initialization and that $v^{(t)}$ is optimal i.e. $v^{(t)} = a^{(t)} w^*$ where $a^{(t)} \gg 1$. Then, the predictor $g$ we would have is

$$g(X) = \sum_{i=1}^{D} \sum_{j=1}^{D} S_{i,j}^{(0)} \langle v^{(t)}, X_j \rangle \propto \sum_{i=1}^{D} \sum_{j=1}^{D} e^{A_{i,j}^{(0)}} \langle w^*, X_j \rangle \tag{2}$$

Such predictor $g$ would yield high population loss because there many more data with random labels ($qD = \text{poly}(C)$) than with the exact label. Therefore, $A_{i,j}^{(t)}$'s start to update. The gradient increment for $\gamma^{(t)}$ (which corresponds to $i$ and $j$ in the same set $\mathcal{S}_\ell$) is much larger than the one for $\rho^{(t)}$ (Lemma 4.3). Thus, $\gamma^{(t)}$ increases until a time $\mathcal{T}_1 \in [\mathcal{T}_0, T]$ such that $\gamma^{(\mathcal{T}_1)} > \max_{t \in [T]} |\rho^{(t)}|$.

- **Event III**: Because we have $\gamma^{(\mathcal{T}_1)} > \max_{t \in [T]} |\rho^{(t)}|$, we again have $\alpha^{(t+1)} > \alpha^{(t)}$ as in Phase I (Lemma D.11). Thus, $\alpha^{(t)}$ increases again until the population risk becomes a $o(1)$.

**Main insights of our analysis.** Our mechanism highlights two important aspects that are proper to attention models:

- because of the initialization and the data structure, we have patch association for any time $t$ (Lemma 4.3).
- our ViT model uses patch association to minimize the population loss (Event III). Without patch association, the model would only be a generalized linear model that does not minimize the loss.

### 4.2 From the idealized to the realistic learning process

The real learning process differs from the idealized one in that we have a finite number of samples and we initialize both $\widehat{A}$ and $\widehat{v}$ as Gaussian random variables. Using a polynomial number of samples, we show that (T) still learns patch association and generalizes.

**Theorem 4.2.** *Assume that we run GD on (E) for $T$ iterations with parameters set as in Parametrization 3.1. Assume that the number of samples is $N = \text{poly}(d)$. With high probability, the model*

1. *learns patch association i.e. for all $\ell \in [L]$ and $i \in \mathcal{S}_\ell$, $\text{Top}_C \{\widehat{A}_{i,j}^{(T)}\}_{j=1}^{D} = \mathcal{S}_\ell$.*

2. *fits the labeling function i.e. $\mathbb{P}_{\mathcal{D}}[f^*(X) F_{\widehat{A}^{(T)}, \widehat{v}^{(T)}}(X) > 0] \geqslant 1 - o(1)$.*

Similarly to [46], the proof introduces a "semi-realistic" learning process that is a mid-point between the idealized and realistic processes. We show that $\widehat{A}^{(T)}$ and $\widehat{v}^{(T)}$ are close to their semi-realistic counterparts – see Appendix E for a complete proof. Figure 4 numerically illustrates Theorem 4.2.

## 5 Patch association yields sample-efficient fine-tuning with ViTs

A fundamental byproduct of our theory is that after pre-training on a dataset sampled from $\mathcal{D}$, our model (T) sample-efficiently transfers to datasets that are structured as $\mathcal{D}$ but differ in their features.

**Downstream dataset.** Let $\widetilde{\mathcal{D}}$ a downstream data distribution defined as in Assumption 1 such that its underlying feature is $\widetilde{w}^*$ with $\|\widetilde{w}^*\|_2 = 1$ and $\widetilde{w}^*$ potentially different from $w^*$. In other words, the downstream $\widetilde{\mathcal{D}}$ and source $\mathcal{D}$ distributions share the same structure but not necessarily the same feature. We sample a downstream dataset $\widetilde{\mathcal{Z}} = \{(\widetilde{X}[i], \widetilde{y}[i])\}_{i=1}^{\widetilde{N}}$ from $\widetilde{\mathcal{D}}$.

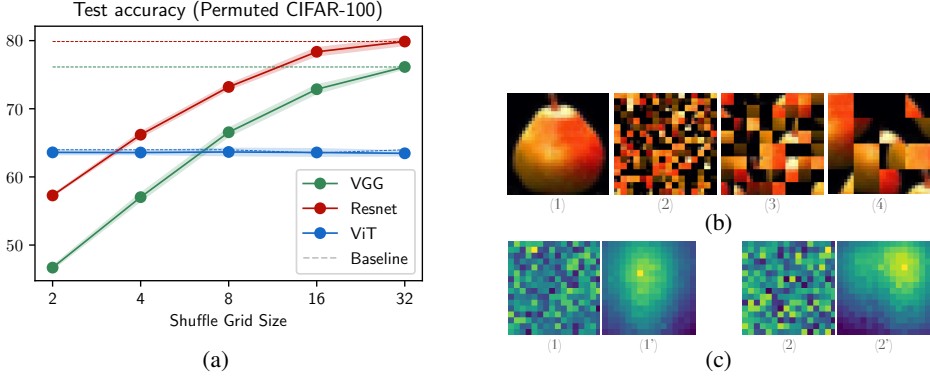

Figure 5: (a): Test accuracy obtained with ViT (patch size 2), ResNet-18 and VGG-19 on permuted (in solid lines) and on original (in dashed lines) CIFAR-100. While convolutional models are very sensitive to permutations, the ViT performs equally whether the dataset is permuted or not. (b): (2) CIFAR-100 image (1) and Permuted CIFAR-100 image when shuffle grid size is 2 (2), 4 (3) and 8 (4). (c): (1-2) Visualization of positional encoding similarities after training a ViT (patch size 2) on permuted CIFAR-100 (shuffle grid size 2). Here, we display $\boldsymbol{p}_i^\top \boldsymbol{P}$ where $i$ is some fixed index and reshape such vector into a matrix $16 \times 16$. We observe that these similarities (1-2) do not have any spatially localized structure. However, when applying the inverse of the permutation, we recover spatially localized patterns in (1'-2').

**Learning problem.** We consider the model (T) pre-trained as in subsection 4.2. We assume that $\widehat{\boldsymbol{A}}$ is kept *fixed* from the pre-trained model and we only optimize the value vector $\widetilde{\boldsymbol{v}}$ to solve:

$$\min_{\widetilde{\boldsymbol{v}}} \quad \frac{1}{\widetilde{N}} \sum_{i=1}^{\widetilde{N}} \log\left(1 + e^{-\tilde{y}[i]F(\widetilde{\boldsymbol{X}}[i])}\right) := \widetilde{\mathcal{L}}(\widetilde{\boldsymbol{v}}). \tag{$\widetilde{\text{E}}$}$$

We run GD on ($\widetilde{\text{E}}$) with parameters set as in Parametrization 3.1 except that the $\widehat{A}_{i,j}$'s are fixed and $\widetilde{\boldsymbol{v}}^{(0)} \sim \mathcal{N}(0, \omega^2 \mathbf{I}_d)$ with $\omega = 1/\text{poly}(d)$. Our main results states that this fine-tuning procedure requires a few samples to achieve high test accuracy in $\widetilde{\mathcal{D}}$. In contrast, any algorithm without patch association needs a large number of samples to generalize.

**Theorem 5.1.** *Let $\widehat{\boldsymbol{A}}$ be the attention matrix obtained after pre-training as in subsection 4.2. Assume that we run GD for $T$ iterations on ($\widetilde{\text{E}}$) to fine-tune the value vector. Using $\widetilde{N} \leqslant \text{polylog}(D)$ samples, the model (T) transfers to $\widetilde{\mathcal{D}}$ i.e. $\mathbb{P}_{\widetilde{\mathcal{D}}}[f^*(\boldsymbol{X})F_{\widehat{\boldsymbol{A}}, \widetilde{\boldsymbol{v}}^{(T)}}(\boldsymbol{X}) > 0] \geqslant 1 - o(1)$.*

**Theorem 5.2.** *Let $\mathcal{A} \colon \mathbb{R}^{d \times D} \to \{-1, 1\}$ be a binary classification algorithm without patch association knowledge. Then, it needs $D^{\Omega(1)}$ training samples to get test error $\leqslant o(1)$ on $\widetilde{\mathcal{D}}$.*

The proofs of Theorem 5.1 and Theorem 5.2 are in Appendix F. These theorems hightlight that learning patch association is required for efficient transfer. We believe that they offer a new perspective on explaining why ViTs are widely used in transferring to downstream tasks. While it is possible that ViTs learn shared (with the downstream dataset) features during pretraining, our theory hints that learning the inductive bias of the labeling function is also central for transfer.

## 6    Numerical experiments

In this section, we first empirically verify that ViTs learn patch association while minizizing their training loss. We then numerically show that the positional attention mechanism competes with the vanilla one on small-scale datasets such as CIFAR-10/100 [43], SVHN [52] and large-scale ones such as ILSVRC-2012 ImageNet [22]. For the small datasets, we use a ViT with 7 layers, 12 heads and hidden/MLP dimension 384. For ImageNet, we train a "ViT-tiny-patch16-224" [24]. Both models are trained with standard augmentations techniques [18] and using AdamW with a cosine learning rate scheduler. We run all the experiments for 300 epochs, with batch size 1024 for Imagenet and 128 otherwise and average our results over 5 seeds. We refer to Appendix A for the training details.

**ViTs learn patch association.** We consider the CIFAR-100 dataset where we divide each image into grids of size $s \times s$ pixels. For a fixed $s \in \{2, 4, 8, 16, 32\}$, we permute the grids according to $\pi_s$ to create the permuted CIFAR-100 dataset. We call $s$ the grid shuffle size. Figure 5b-(1) shows a CIFAR-100 image and its corresponding shuffling in the permuted CIFAR-100 dataset Figure 5b-(2-3-4). We train a ViT and CNNs ResNet18 [31] and VGG-19 [57] on the permuted CIFAR-100 dataset.

For the ViT, we set the patch size to 2, although this is sub-optimal in terms of accuracy, because the patch size needs to stay smaller or equal to $s$. Indeed, intuitively, when we permute the grids in Figure 5b, we lose the local aspect of the spatial structure and create new sets $\mathcal{S}_\ell$'s and a new labeling function $f^*$. Figure 5a reports the test accuracy of these three models for different values of $s$. When $s$ is small, the image does not have a coherent structure e.g. Figure 5b-(2) and thus, CNNs struggle to generalize. As $s$ increases e.g. Figure 5b-(4), the information inside a patch is

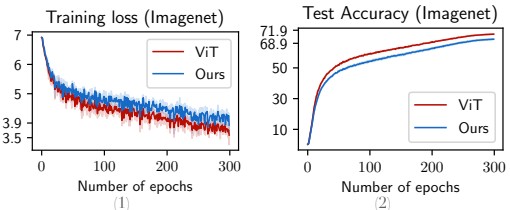

Figure 6: Training loss (1) and test accuracy (2) obtained using a ViT-tiny-patch16-224 on Imagenet. ViT using positional attention (Ours) gets 68.9% test accuracy while vanilla ViT (ViT) gets 71.9%.

meaningful and thus, the CNNs well-perform. Unsurprisingly, since ViTs are *permutation invariant*, their performance remains unchanged for all $s$ – see Figure 5a. Despite this change, we verify that the ViT is able to recover the new $\mathcal{S}_\ell$'s: we feed the ViT with the shuffled pear image (Figure 5b-(2)) and consider for some $i$ the similarity matrix $p_i^\top P$. We see that it does not exhibit a local spatial structure in Figure 5c-(1,2). We then apply $\pi_s^{-1}$ to $p_i^\top P$ and observe that we recover the spatially localized patterns Figure 5c-(1',2'). This experiment highlights that ViTs do not just group nearby pixels together as convolutions. They learn a more general spatial structure, in accordance to our theoretical results.

**ViTs with positional attention are competitive.** We numerically verify that ViTs using positional attention compete with those with vanilla attention. In Section 3, we introduced positional attention to define our theoretical learner model. Figure 6 and Figure 7 show that ViTs using positional attention compete with vanilla ViTs on a range of datasets. These experiments strengthen our intuition that for images, having an attention matrix that only depends on the positional encodings is sufficient to have a good test accuracy.

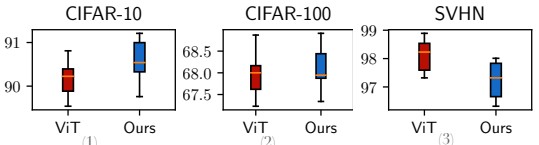

Figure 7: Test accuracy obtained with a ViT using vanilla attention (ViT) and positional attention (Ours) on CIFAR-10 (1), CIFAR-100 (2) and SVHN (3). Our model competes with the vanilla ViT. Patch size 4 and average over 10 seeds for this experiment.

## Conclusion, limitations and future works

Our work is a first step towards understanding how Transformers learn tailored inductive biases when trained with gradient descent. Our analysis heavily relies on the positional attention mechanism that disentangles patches and positional encodings. In practice, self-attention mixes these two quantities. An interesting direction is to understand the impact of patch embeddings on the inductive bias learned by ViTs. Moreover, our experiment on the Gaussian data shows that ViTs do not always learn the correct inductive bias under Definition 2.1: characterizing the distributions under which ViTs recover the structure of the function is an important question. Lastly, this work also paves the way to many extensions beyond convolution. For example, can ViTs learn other inductive biases? What are the inductive biases learnt by Transformers in NLP? Answering those questions is central to better understand the underlying mechanism of attention.

## Acknowledgments and Disclosure of Funding

The authors would like to thank Boris Hanin for helpful discussions and feedback on this work.

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
