# OpenReview forum: "Vision Transformers provably learn spatial structure"
_NeurIPS.cc/2022/Conference — NeurIPS 2022 Accept_

### Official Review · Reviewer_gkYb · 2022-07-07

**Rating:** 4
**Confidence:** 3
**Soundness:** 3 good
**Presentation:** 2 fair
**Contribution:** 2 fair

**Summary:**

Vision Transformers

The paper focuses on trying to explain how Vision Transformers learn the spatial locality biases on images. To do so, authors mathematically analyze the learning of convolution-like structures on ViT and test their idea with some synthetic datasets based on CIFAR.



**Questions:**

- The test of L116-129 is not so clear to me. Where is the cubic activation applied? Why 1 layer? Would you have the same result with more layers? not clear to me
- Why do we need to learn so well how positional encodings are learned?

**Limitations:**

I think the authors do not have major limitations or negative societal impact.

**Strengths And Weaknesses:**

PROS:
- the paper seem to be very detailed and provides a mathematical analysis of ViT

CONS:
- very hard to read
- limited/niche contribution

DETAILS:
Generally speaking, this paper is VERY difficult to follow. My background is computer science but I found it hard to read this paper that is FULL of definitions, assumptions, definitions, theorems but, most importantly, the paper introduces many many letters in each section.

The math seems reasonable but I have to admit that around line 200 I was already a bit lost and I could not follow the paper so well. I might have missed something. This raises also a problem for me: I did not understand how ViT learns the spatial inductive bias after reading 9 pages only about that. Moreover, I am not even sure that the tests done with 1-layer ViTs are generalizable to more layers.

Moreover, I think the paper misses one main point: explaining WHY it is important to learn how ViT learns the inductive biases. I know it sounds a stupid comment, but without this explanation a reader is not intrigued by the paper and the heavy math inside it. Then, once we know that “ViT do not just group nearby pixel together” why is it useful? I mean, I am very naive but self-attention is based on content, so I am not surprised that it does not group nearby pixel together.

Here some details from my review:
- L29: not clear. Are you just saying that near pixels are more similar to distant pixels? Because it not so clear that distant pixels encode “abstract" features.
- Figure 1 b should not be there. Better to move it near the text.
- L80 this notation is not a related work, is it?
- L90 at this point is not clear what C is
- L91 why is a definition expected to be an assumption? L90-94 is not clear at all
- The test of L116-129 is not so clear to me. Where is the cubic activation applied? Why 1 layer? Would you have the same result with more layers? not clear to me
- L140 is Threshold_0.9C might be Threshold_0.9
- I would modify the title in something more clear like “How do Vision Transformers learn…"

---

> ### Author Response · Authors · 2022-08-02
> **Response to Reviewer gkYb**
>
> We thank Reviewer gkYb for their comments. We address their concern as follows.
>
> **Explanation of the contributions**
>
>
> Our work attempts to understand  how attention-based models implicitly learn a particular inductive bias while minimizing their training loss. In the case of vision datasets, it has been empirically observed that while the positional encodings (PEs) are initially random (Figure 1 (a) top), the final PE of a fixed token is highly correlated with the ones of his neighbors. This phenomenon occurs **implicitly** while the ViT only minimizes its loss. Understanding how the PE gets this special structure during training may help us understanding why ViTs work and what  operations they are able to learn. While it is known that attention models can express local operations as convolutions [1], it remains unclear how they learn them. To summarize, our main contributions are i) proving that starting from random PEs, an attention-based model can learn correlated PEs as in Figure 1 (a) just by minimizing its training loss with gradient descent and ii) detail the mechanism by which we get this result.
>
> Another main result we show is that after pre-training, our ViT model can be fine-tuned in a sample-efficient manner to downstream datasets that share the same structure but differ in their feature. We believe that this is a novel result in transfer learning since most of the works in these areas assume shared representations between the pre-training and downstream datasets [2,3]. It gives a potential explanation of why ViTs are popular for transfer learning.
>
> **I did not understand how ViT learns the spatial inductive bias**
>
>
> We previously had a paragraph describing the learning mechanism. We modified it in Section 4.1 to highlight the specifics of attention models.
>
>    - **Event 1**: At the beginning, the gradient increment for $v$ is larger than the one of $A_{i,j}^{(t)}$. Because the $A_{i,j}^{(t)}$'s stay very small and  $v^{(t)}$ is initialized in span$(w^*)$, $v$ updates in span$(w^*)$. At this point, the model is a generalized linear model that would not generalize because there are much more noisy tokens than signal ones (Theorem 3.1).
>
>  - **Event 2**: Once $< v^{(t)},w^*>$ becomes large, the $A_{i,j}^{(t)}$'s start to update. At this point, because the $A_{i,j}$'s are not negligible, $v^{(t)}$ updates in the direction of $w^*$ and of $\xi_j$. However, the gradient increment for $\gamma$ is much larger than the one for $\rho$ (Lemma 4.3). Thus, $\gamma^{(t)}$ increases until being large enough so that the gradient of $L$ with respect to $v$ is in  in span$(w^*)$.
>
> - **Event 3**: Since the gradient of $L$ is in the direction of $w^*$, $v$ updates again in span$(w^*)$ until population risk becomes a $\Omega(1)$ and we fit the labelling function.
>
> Our mechanism highlights a main advantage of attention-based models: their ability to perform patch association i.e. to set high $A_{i,j}$ for $i,j$ in the same set $S_l$ By doing so, we show that they are able to align their gradient in span$(w^*)$ and fit the labelling function. Another by-product of this ability is  to have sample-efficient transfer as highlighted in Section 6.
>
>
>
> [1] Cordonnier, Jean-Baptiste, Andreas Loukas, and Martin Jaggi. "On the relationship between self-attention and convolutional layers." arXiv preprint arXiv:1911.03584 (2019).
>
> [2] Tripuraneni, Nilesh, Michael Jordan, and Chi Jin. "On the theory of transfer learning: The importance of task diversity." Advances in Neural Information Processing Systems 33 (2020): 7852-7862.
>
> [3] Baxter, Jonathan. "A model of inductive bias learning." Journal of artificial intelligence research 12 (2000): 149-198.

---

### Official Review · Reviewer_4CvW · 2022-07-12

**Rating:** 5
**Confidence:** 4
**Soundness:** 3 good
**Presentation:** 4 excellent
**Contribution:** 2 fair

**Summary:**

As observed by the authors, the advent of the vision transformer architecture is surprising given that their network structure has nothing specifically curated for vision problems. This raises the question of how it is that, in training a vision transformer, certain desirable properties for vision problems are obtained in the resulting models.  In the abstract authors refer to these as "convolutional-like patterns", by which they mean the patches of the image are embedded with high cosine similarity to adjacent patches. The question the authors seek to address is how it is that there resulting positional encodings are learnt.

The model considered is called by the authors "positional attention" which, if I understand correctly, consists of a single self-attention layer where only the positional embeddings are fed into the attention mechanism when computing the attention matrix. In this setting, when the query and key matrices are held fixed as identities (Assumption 3.1), the one-layer model learns to associate patches in a manner consistent with the assumed data generating process, (1), i.e. Theorem 4.1.

**Questions:**

* What is missing from this paper are two things, one which may be more easily addressed than the other:
	1. what the relationship is between this simplified structured partitioning problem and and the real phenomenon of local cosine similarity among positional embeddings in a vision transformer; and
	2. what these results tell us about the "*how*" portion of the question posed above.

* Supposing (1) is answered, the results in the paper appear merely to prove that the phenomenon happens, rather than answering questions about how or why it happens.

* The closest insight to (2) this appears to be Theorem 3.1 which says that GLM's cannot solve these structured problems. Okay, so we must need something more complicated than a GLM? Is it a ViT? is it a single layer positional attention model? Is it something else we haven't even thought of?

* By removing the inputs from the attention mechanism, is it really fair to call the resulting architecture "attention" at all?

* The conclusion reads *"Our work is a first step towards understanding how Transformers learn tailored inductive biases when trained on large datasets with gradient descent."* What exactly the first that first step? What new understanding should I take away from this work about the behaviour of vision transformers.


**Limitations:**

* The work would be strengthened by an analysis of the limitation of the data generating process assumption. Since this is not the way labels yielded conventionally for vision transformers. I.e. in the case of the ViT the features were extracted from the `[pad]` token output.

* Much of the discussion about "competitiveness" of "positional attention" seems like it would have benefitted by an analysis looking at how much of the error gradient flowed to the original input matrix $X$ in the sum $X+P$ in the pre-transformer layer. In the sense that if really it makes no difference to tweak the architecture in this fashion then $\mathrm{SA}(X;P) \approxeq \mathrm{PA}(X; P)$. Really the point here should be to argue for the modifications to the architecture for positional attention, which I am not observing.

* Still the more fundamental question about translation invariance of convolution seems to be have forgotten despite "convolution" appearing throughout.

* Given the implications and assumptions made here, and the claim that this is a first step towards understanding something, what are the possible next steps? What is the next step get more of a handle on features of the transformer architecture that either make it unique, or ordinary?

**Strengths And Weaknesses:**

I will preface this section by observing that I believe the term "convolutional pattern" to be a misnomer and misleading. The defining quality of the convolutional layer is translation invariance. What the authors study instead is how transformers learn positional embeddings with high local cosine similarity; which is quite different from a more interesting question which is how transformer models adapt to domains where convolutional architectures have remained dominant. For my review I will use the terminology consistent with the paper. The confusion extends to the description of the partitions $(\mathcal{S}_\ell)_\ell$ being "convolutional".

#### Strengths
* While I have not examined the supplementary material in detail, the authors have a technical work with nontrivial results which I applaud.

* The paper is well written with good, clear notation.

* In Section 3 that the authors make an effort to show the nontrivially of the data generating process in sense that it cannot be fit with low error by a GLM. (Theorem 3.1) . I.e. learning the positional embeddings is necessary generalisation given the data generating process.

* The permutation experiments in Section 4 are interesting.

#### Weaknesses
The biggest weakness for this work I feel is the overall style. The original question posed by the authors "****how**** do ViTs learn [convolutional patterns] by solely minimizing their training loss?" is an interesting one. However it is not clear to me how the (well written) theorems and results shed light on this question.



#### Typographic:
- [90] $\mathcal{S}_\ell  \subsetneq  D$ should read $\mathcal{S}_\ell  \subsetneq  [D]$ and a quantifier over $\ell$ should be inserted: "for all $\ell \in [L]$"
- [195] Observe your hyperlink to "Assumption 3.1" is linking to a theoremlike environment titled "Simplification 3.1"

---

> ### Author Response · Authors · 2022-08-02
> **Response to Reviewer 4CvW**
>
> We thank Reviewer 1xfl for their very good review. We took into account several of his comments and integrated the changes they asked in the paper. We address here the most crucial questions they ask.
>
> - **Insights on how does the model learn the  real phenomenon of local cosine similarity  + on how positional attention compares with self attention"**:
>
> We invite the reviewer to look at the general response to all the reviewers that is titled "Summary of the reviews" for our answers to this point.
>
> - **"Relationship between  the simplified structured partitioning problem and the real phenomenon of local cosine similarity"**
>
> We added a paragraph in the beginning of Section 3 to address this point. In Figure 1 (a) bottom, we observe that given a patch $i$, its PE is highly correlated with those of his neighbors.
> In order to prove this empirical phenomenon, we need a data model. We propose a simplified model where each $X_j$ can be thought as a patch and the $\mathcal{S}_{\ell}$'s are sets gathering contiguous patches. Therefore,  in our setting, observing Figure 1 (a) bottom is equivalent to having for token $i$, its PE highly correlated with the PEs of patches in
>
> $\mathcal{S}_{\ell}$. Mathematically, it is equivalent to for all $i\in [D]$,
>
> $Top_C< p_i, p_j> $ $=\mathcal{S}_{\ell}$.
>
> - **How do the theorems answer the question "how dot ViTs...?**
>
> Our main result (Theorem 4.2) shows that by just minimizing the training loss, our model i) generalizes ii) starting from random PEs, we end up with PEs that satisfy: for all $i\in [D]$,
>
> $\mathrm{Top}_{C}\langle p_i, p_j\rangle =\mathcal{S}_{\ell}$ which is a math description of what we observe in Figure 1 a bottom. Thus Theorem 4.2. is a solution to the question we have set.
>
> - **Confusion around "convolution-like patterns"**
>
> We apologize for the confusion around "convolutional pattern". What we meant by this term is the fact that the ViT is able to group nearby tokens like in Figure 1 or Figure 5. We change "convolutional pattern" by "localized patterns" in our updated version of the paper.

---

> > ### Comment · Reviewer_4CvW · 2022-08-10
> > **Post Rebuttal Response**
> >
> > Thank you for addressing my concerns. I appreciate the changes you have proposed.
> >
> > After consideration, as an editorial matter, I think couching your results in terms convolution is really doing you a lot more harm than good. However, the ability to learn the ground-truth associativity in the positional embeddings is a nontrivial result that should be published.
> >
> > If I may make a suggestion, I believe one of two possible further revisions needs to be made:
> > 1. (you are able to) demonstrate a clear relationship between a translational-invariant (convolutional) layer and the ground-truth positional embeddings, for some distribution, for some loss; or
> > 2. (easier) rework the paper to not couch the results with *any* reference to convolution.
> >
> > If you are able to accomplish either of these I will be happy to revise my score and recommend your paper for inclusion to the area chair. I realise that we are nearing the end of the response period, but if you agree please let either me or the area chair know.

---

### Official Review · Reviewer_1xfL · 2022-07-13

**Rating:** 5
**Confidence:** 4
**Soundness:** 2 fair
**Presentation:** 3 good
**Contribution:** 2 fair

**Summary:**

The paper provides a theoretical analysis of gradient descent on logistic loss under simplified formulations of model and data (represented as a collection of $D$ tokens/patches). The assumptions and their relevance are detailed below, but briefly at high level the data assumption models presence of structured association between patches, and the model captures a (significantly) simplified attention mechanism that is loosely motivated from transformer models.
The key result shows that gradient descent in a range of hyperparameters learn the patch associations under the model and data assumptions (more details below).


**Questions:**

Technical questions:

In proof of Lemma 4.2 in Appendix I.1, I believe the second line in eq. in 205 does not hold $\xi$ also appears as part of $X$ in other terms. To elaborate, from Definition 3.1, X depends on samples of $\xi_j, \delta_j, l(X)$, so the expectation is of the form $E_{\xi,l(X),\delta_j) g(X)X$ cannot in general be split as $E[g(X)E_\xi[X]]$. From briefly checking the options, I think we need $\xi_j=0$ as assumption for he lemma to go through.

Other comments:

(a)	In the context of the analysis and results in the paper, it’s unclear what the significance of the the Gaussian noise $\xi_j$ in the data model in Defn. 3.1. From my understanding of the proof, although the noise technically captures a more general model, it does not provide any new insight into the analysis/results itself and merely leads to complicated bookkeeping. If I had missed their significance, I request the authors to clarify in their response. If not, having a simpler analysis without the noise as the main result would improve the readability of the paper significantly (the noisy general analysis could be deferred to separate appendix section).

(b)	Definition 2.3 might need slight modification of wording to ensure there are no-tie breaking required for top_C, for example for a model that learns all position encoding as the same, say $p_i=p$, whether it learns patches or not according to this definition depends on how we do the tie breaking.

(c)	Definition 2.1: where each *patch* X -> *input* X

(d)	Fig 2: what is the kernel size of f^*? Also, it is unclear what exactly is being plotted in UpperRight in the  DxD matrix – if it is the ideal convolutional structure, why is it not all 1 in the diagonally entries? For LowerRight, what is the normalization used (not that P^\top P  is not intrinsically normalized)

(e)	One way to rephrase the insight of Fig 2. Is that the patch learning may not happen when merely $f^*(X)=y|X$ has a patch structure, but rather we might require conditions on marginals of distribution over X to have patch structure – this might have been obvious to the authors and experts, but I believe some readers might benefit from explicitly stating this intuition.


**Limitations:**

Yes

**Strengths And Weaknesses:**

The main results to my knowledge are correct except for the point under “Technical questions” below – although I did not check all the proof details. This work is in the line of results that aim to identify interesting inductive biases of models learned using gradient descent. Modern neural networks are extremely complex to theoretically capture in full generality and inevitably any mathematical analysis would rely on significant simplifications. My evaluation of the paper relies on to what extent the paper provides new ideas/insights into the workings of attention based models and I discuss the strengths and weaknesses of the assumptions below.

*Data assumption:* The data distribution in Definition 3.1 captures scenarios where learning patch/token associations are important, in a sense that the labels as well as marginals depend on certain grouping of tokens. In a nutshell, the $D$ data patches are grouped into L disjoined sets ${S_l:l=1,2,…,L}$ where each set has $C$ patches, with $C<<D$. The goal is to understand when a model/algorithm can learn the patch association/groupings $S_l$.  The dimension scaling in the data model captures high SNR scenarios with spurious features.
This is a good simple data model to theoretically understanding of learning structures present in data and is of interest beyond this paper. Secondary results in the paper establish that this model is *not* learnable by any generalized linear model, and moreover it is possible for a one-layer transformer model can represent the labeling distribution even without learning the patch association structure (Thm 3.1 & 3.2).

*Model and hyperparameter assumption:* The model on inputs denoted as $X\in\mathbb{R}^{d\times D}$ that is studied by the paper can be written as
$$ F(X)=\sigma(\langle X, v\cdot s_i^\top\rangle),\text{ where } s_i=softmax(a_i)$$
where $a_i\in\mathbb{R}^D$ for $i=1,2,\ldots, D$ and $v\in\mathbb{R}^d$ are unconstrained parameters over which gradient descent is performed, and $\sigma$ is polynomial activation. $S,A$ are $D\times D$ matrices with ${s_i},{a_i}$ along the rows.

In my evaluation, the main shortcoming of the paper is that the simplifications in model and hyperparameter assumptions do not capture aspects of transformer/attention-based models to a sufficiently satisfactory extent. The authors acknowledge that in their paper in multiple places and I am not opposed extreme simplifications if they capture some new insight about the models the paper aims to address. Unfortunately, with the combination of above modeling choice, it is not clear what aspects of attention model are still being captured.

1.	The motivation for the formulation is based loosely on transformers, where A and S correspond to attention and score matrices. However, the connection is very loose in that A does not depend on the inputs at all.  Even with the simplification that the attentions only depend on the positional encoding (which I think is reasonable for image tasks and is also justified with experiments), the standard form would involve parametrizing $A$ in factorized form as $P^\top W_QW_K^\top P$ -- even though the factorized form does not change representation, it is certainly known to change the gradient descent dynamics which is not captured in this form. As stated in the above equation, it is unclear how the model even distinguishes attention mechanism from any other simplification of fully connected networks.

2.	A more important concern for me arises from the choice of initialization hyperparameters. With initialization $a_{ii}=polyloglog(d)$ and $a_{ij}=1/poly(d) $. This provides a strong prior on initial score matrices as $s_ii\approx 1-o(1)$ and $s_ij=O(1/d)$ -- i.e., the scores are explicitly chosen to be diagonally dominant. This in comparison to even the simplified parametrization as $A=P^\top W_QW_K^\top P$ would lead to a more balanced scores at initialization—thus the initialization is truly agnostic to structure and the mechanism if learns patch association at all will have to learn from scratch.

---

> ### Author Response · Authors · 2022-08-02
> **Response to Reviewer 1xfl**
>
> We thank Reviewer 1xfl for their  very good review. We took into account several of his comments and integrated the changes they asked in the paper. We address here the most crucial questions they ask.
>
> - **Insights on the mechanism + on how positional attention compares with self attention"**:
>
> We invite the reviewer to look at the general response to all the reviewers that is titled "Summary of the reviews" for our answers to this point.
>
> - **Factorized form for the attention matrix**
>
> We agree with the reviewer that from an optimization perspective, the factorized form is not equivalent to the non-factorized one. However, we decide to opt for the non-factorized form for two reasons: i) we want the simplest attention-based model for our analysis to show that even with stringent assumptions, we are able to prove implicit inductive bias learning. ii) as stated by the reviewer and mentioned in the paper, the factorized and non-factorized forms have the same representation power. Understanding the dynamics of the factorized form and comparing it with our model is an exciting direction.
>
> - **Choice of the initialization in the idealized case**
>
> We included a paragraph after Parametrization 3.1 to clarify our initialization choice.
> We first remind that $A=P^{\top}P$ and each entry in this matrix is $\langle p_i,p_j\rangle.$ Assume that we sample the PEs $\{p_1,\dots,p_D\}$ as i.i.d.\ Gaussian random variables in dimension $d' = \mathrm{poly}(d)$ of variance $\mathrm{polylog}(d)/d'.$ Then, by high concentration, we have the diagonal of A as $\|p_i\|_2^2=\Theta(\mathrm{polylog}(d))$ and the off diagonal terms as $|\langle p_i,p_j\rangle| =\Theta(\mathrm{polylog}(d)/\mathrm{poly}(d)).$ To summarize, our initialization in Parametrization 3.1 is an analogy to a random initialization of the PEs.
>
> - **Proof of Lemma 4.2 in Appendix I**
>
> We apologize there is indeed a typo in the proof that we corrected in the updated version of the paper. The expectation is taken with respect to $\xi|_{\mathrm{span}(w^*)^{\perp}}$ and not $\xi$. We detail how we compute this expectation. The noise is present in the dot products $\langle v, X_j\rangle$. In the idealized case, we have the initialization $v=\alpha w^*$ which implies that
>
> $< v, X_j> =< \alpha w^*, \delta_j w^*+ \xi|_{\mathrm{span}(w^*)} w^{*}$  $+ \xi|_{\mathrm{span}(w^*)^{\perp}}  u  >   $
>
> $ = \alpha ( \delta_j +\xi|_{\mathrm{span}(w^*)} )$ where $u\in\mathrm{span}^{\perp}(w^*).$ Thus, $\langle v,X_j\rangle$
>
> does not depend on  $ \xi|_{\mathrm{span}(w^*)^{\perp}} $
>
> and we can take directly take the expectation $E_{\xi|_{\mathrm{span}(w^*)^{\perp}}}[X_j]$ in (206).
>
> - **Role of noise in the analysis**
>
> In our analysis, the feature noise plays an important role. Indeed, without it, generalized linear models would be able to generalize in our setting (see the explanation above). We need the noise to show that the attention-model is forced to learn patch association to eventually generalize.
>
> - ** Figure 2: what is the kernel size of f^*? Also, it is unclear what exactly is being plotted in UpperRight in the DxD matrix – if it is the ideal convolutional structure, why is it not all 1 in the diagonally entries? For LowerRight, what is the normalization used (not that P^\top P is not intrinsically normalized)**
>
> The kernel size is 2.  We apologize, the upper-right figure is indeed confusing. We updated it in the revised paper to make it more understandable. Our labelling function is a convolutional network. What we display here are the different sets S_l's that are defined in Assumption 1. In our plot, a patch (i,j) is in yellow if it is taken into account when computing the labelling function and is blue otherwise.  We thus retrieve the ones on the diagonal as expected by the reviewer.
>
> We didn't use any normalization for the lower right figure. Our goal was simply to show that we could not see the spatially localized patterns with the ViT model.
>
> - **Difference of our model with MLPs and CNNs**
>
> We now clarify how our model compares to MLPs and CNNs. As all attention models, MLPs can express our simplified ViT model. However, our model is more general and includes CNNs. Indeed, a useful analogy is  to think of the attention matrix as a filter and the value matrix as the input. In our model, we have a filter that depends on the position (and not on the input value) of the patch similarly to CNNs. However, contrary to CNNs, our filter simultaneously applies to all the patches (and not to a spatially localized patch). Therefore, the main difference between our model and a CNN is the fact that our filter is "delocalized". This aspect  is important for instance in settings where convolution-based models fail e.g. when we permute all the tokens (see Figure 5).

---

### Official Review · Reviewer_7w7a · 2022-07-15

**Rating:** 4
**Confidence:** 3
**Soundness:** 2 fair
**Presentation:** 2 fair
**Contribution:** 2 fair

**Summary:**

This work aims to understand how Transformers learn inductive biases as convolution when trained on large datasets with gradient descent. They provide detailed analysis and find that ViT relies on the positional attention mechanism that disentangles patches and positional encodings. And it also provides experimentS on the Gaussian data that show that ViTs do not learn the correct inductive bias under the assumption that "characterizing the distributions under which ViTs recover the structure of the function is an important question." Lastly, it also provides experiments on CIFAR and ImageNet.

**Questions:**

Please refer to Weaknesses.

**Limitations:**

Please refer to Weaknesses.

**Strengths And Weaknesses:**

Strengths:

1. The analysis of this work is completed and somehow interesting.
2. The definition and theorem on the patch association are good.


Weaknesses:
1. Even though the analysis is complete, the final results are not surprising and the improvement on downstream tasks is incremental. I think good work should be simple but effective or complex but effective. This work does not give me any new insight.
2. In Figure 1(a), why does the author state that ViTs learn the convolution inductive bias in vision datasets? I do not see any convolution inductive bias in this figure.

---

> ### Author Response · Authors · 2022-08-02
> **Response to Reviewer 7w7a**
>
> We thank Reviewer 7w7a for their comments. We address their concerns as follows:
>
>  - **the final results are not surprising and the improvement on downstream tasks is incremental.**
>
>
> Our work attempts to understand  how attention-based models implicitly learn a particular inductive bias while minimizing their training loss. In the case of vision datasets, it has been empirically observed that while the positional encodings (PEs) are initially random (Figure 1 (a) top), the final PE of a fixed token is highly correlated with the ones of his neighbors. This phenomenon occurs \textbf{implicitly} while the ViT only minimizes its loss. Understanding how the PE gets this special structure during training may help us understanding why ViTs work and what  operations they are able to learn. While it is known that attention models can express local operations as convolutions [1], it remains unclear how they learn them. To summarize, our main contributions are i) proving that starting from random PEs, an attention-based model can learn correlated PEs as in Figure 1 (a) just by minimizing its training loss with gradient descent and ii) detail the mechanism by which we get this result.
>
> Another main result we show is that after pre-training, our ViT model can be fine-tuned in a sample-efficient manner to downstream datasets that share the same structure but differ in their feature. We believe that this is a novel result in transfer learning since most of the works in these areas assume shared representations between the pre-training and downstream datasets [2,3]. It gives a potential explanation of why ViTs are popular for transfer learning.
>
> - **why does the author state that ViTs learn the convolution inductive bias in vision datasets?**
>
> We apologize for the confusion around "convolutional pattern". In our revised version of the paper,  we change "convolutional pattern" by "spatially localized patterns". What we meant by this term is the fact that the ViT is able to group nearby tokens like in Figure 1 or Figure 5.
>
> - **This work does not give me any new insight**
>
> We previously had a paragraph describing the learning mechanism. We modified it in Section 4.1 to highlight the specifics of attention models.
>
>    - **Event 1**: At the beginning, the gradient increment for $v$ is larger than the one of $A_{i,j}^{(t)}$. Because the $A_{i,j}^{(t)}$'s stay very small and  $v^{(t)}$ is initialized in span$(w^*)$, $v$ updates in span$(w^*)$. At this point, the model is a generalized linear model that would not generalize because there are much more noisy tokens than signal ones (Theorem 3.1).
>
>  - **Event 2**: Once $< v^{(t)},w^*>$ becomes large, the $A_{i,j}^{(t)}$'s start to update. At this point, because the $A_{i,j}$'s are not negligible, $v^{(t)}$ updates in the direction of $w^*$ and of $\xi_j$. However, the gradient increment for $\gamma$ is much larger than the one for $\rho$ (Lemma 4.3). Thus, $\gamma^{(t)}$ increases until being large enough so that the gradient of $L$ with respect to $v$ is in  in span$(w^*)$.
>
> - **Event 3**: Since the gradient of $L$ is in the direction of $w^*$, $v$ updates again in span$(w^*)$ until population risk becomes a $\Omega(1)$ and we fit the labelling function.
>
> Our mechanism highlights a main advantage of attention-based models: their ability to perform patch association i.e. to set high $A_{i,j}$ for $i,j$ in the same set $S_l$ By doing so, we show that they are able to align their gradient in span$(w^*)$ and fit the labelling function. Another by-product of this ability is  to have sample-efficient transfer as highlighted in Section 6.
>
>
>
> [1] Cordonnier, Jean-Baptiste, Andreas Loukas, and Martin Jaggi. "On the relationship between self-attention and convolutional layers." arXiv preprint arXiv:1911.03584 (2019).
>
> [2] Tripuraneni, Nilesh, Michael Jordan, and Chi Jin. "On the theory of transfer learning: The importance of task diversity." Advances in Neural Information Processing Systems 33 (2020): 7852-7862.
>
> [3] Baxter, Jonathan. "A model of inductive bias learning." Journal of artificial intelligence research 12 (2000): 149-198.

---

### Author Response · Authors · 2022-08-02
**Summary of the reviews**

We thank the reviewers for their insightful comments. We **took into account many of the comments in our revised version of the paper that we have uploaded during the rebuttal**. To make them more apparent, we made these changes in red. The reviewers  **"applaud" and appreciate our theoretical results** (Reviewers 7w7a and 4CvW) and found our paper **"well-written" with "clear notation"** (Rev. 4CvW).   Rev. 1xfL found our **data setting ``of interest beyond this paper''** and Rev. 1xfL  and 4CvW liked that we **compare** our theory result on transformers **with baselines like generalized linear models.** We address here the main concerns .

- Reviewers  7w7a and gkYb: **"Final results are not surprising", explanation of our contributions**

Our work attempts to understand  how attention-based models implicitly learn a particular inductive bias while minimizing their training loss. In the case of vision datasets, it has been empirically observed that while the positional encodings (PEs) are initially random (Figure 1 (a) top), the final PE of a fixed token is highly correlated with the ones of his neighbors. This phenomenon occurs **implicitly** while the ViT only minimizes its loss. Understanding how the PE gets this special structure during training may help us understanding why ViTs work and what  operations they are able to learn. While it is known that attention models can express local operations as convolutions [1], it remains unclear how they learn them. To summarize, our main contributions are i) proving that starting from random PEs, an attention-based model can learn correlated PEs as in Figure 1 (a) just by minimizing its training loss with gradient descent and ii) detail the mechanism by which we get this result.

- Reviewers 1xfL and 4CvW: **"What aspects of attention-based models are captured [by our model]?"**

We added to the paper a paragraph in Section 3 where we address this concern.
We first argue that our PA is an attention mechanism in that we compute a score matrix where rows are equal to 1 and use it to multiply with a value vector.

The question is rather about the aspects of SA captured by PA. We remind that our goal is to prove that starting from random PEs, ViTs generalizes and implicitly learns PEs as  in Figure 1a.  Given its non-linear nature with respect to the inputs, directly analyzing SA is difficult. Thus, **for simplicity**,  we opt for a model where PEs and inputs  are disentangled: the attention matrix solely encodes PEs and the value matrix the amount of learnt feature.  Thus, the attention matrix is used to prove the phenomenon in Figure 1 and the generalization aspect is encoded in $v$. Besides, we believe that PA encodes most of the inductive bias present in SA for **computer vision** datasets. Indeed, as shown in Figure 6, we show that training a ViT on Imagenet with PA results in a tiny accuracy deterioration compared to SA (~3%). Lastly, to our knowledge, there is no work analyzing the GD dynamics of attention-based models.

To summarize, the main insights our PA model captures from SA are i) PA is permutation-invariant and processes all the tokens at the same time ii) PA also has the softmax weights that allows to have a sparse score matrix.

- Reviewers 1xfL and 4CvW: **Specific insights of attention models on the mechanism**

We previously had a paragraph describing the learning mechanism. We modified it in Section 4.1 to highlight the specifics of attention models.

   - **Event 1**: At the beginning, the gradient increment for $v$ is larger than the one of $A_{i,j}^{(t)}$. Because the $A_{i,j}^{(t)}$'s stay very small and  $v^{(t)}$ is initialized in span$(w^*)$, $v$ updates in span$(w^*)$. At this point, the model is a generalized linear model that would not generalize because there are much more noisy tokens than signal ones (Theorem 3.1).

 - **Event 2**: Once $< v^{(t)},w^*>$ becomes large, the $A_{i,j}^{(t)}$'s start to update. At this point, because the $A_{i,j}$'s are not negligible, $v^{(t)}$ updates in the direction of $w^*$ and of $\xi_j$. However, the gradient increment for $\gamma$ is much larger than the one for $\rho$ (Lemma 4.3). Thus, $\gamma^{(t)}$ increases until being large enough so that the gradient of $L$ with respect to $v$ is in  in span$(w^*)$.

- **Event 3**: Since the gradient of $L$ is in the direction of $w^*$, $v$ updates again in span$(w^*)$ until population risk becomes a $\Omega(1)$ and we fit the labelling function.

Our mechanism highlights a main advantage of attention-based models: their ability to perform patch association i.e. to set high $A_{i,j}$ for $i,j$ in the same set $S_l$ By doing so, we show that they are able to align their gradient in span$(w^*)$ and fit the labelling function. Another by-product of this ability is  to have sample-efficient transfer as highlighted in Section 6.


[1] Cordonnier, J-B, Loukas, A, Jaggi, M. "On the relationship between self-attention and convolutional layers."(2019).

---

### Meta-Review · Area_Chair_yr4v · 2022-08-21

**Recommendation:** Accept
**Confidence:** Less certain

**Metareview:**

This paper provides a theoretical analysis of the empirical finding that Vision Transformers learn position embeddings that recapitulate the spatial structure of the training data, even though this spatial structure is no longer explicitly represented after the image is split into patches. The reviewers are generally satisfied by the soundness of the theory, but there is some disagreement regarding the significance of the contribution. The AC believes this paper asks an interesting theoretical question, even if (as is often true) it can only be answered in a simplified setting, and the answer is nontrivial. The AC thus recommends acceptance.

**Award:**

No

---

### Decision · Program_Chairs · 2022-09-14

Accept